# Application of CCTV Methodology to Analyze COVID-19 Evolution in Italy

**DOI:** 10.3390/biotech11030033

**Published:** 2022-08-11

**Authors:** Marianna Milano, Giuseppe Agapito, Mario Cannataro

**Affiliations:** 1Department of Medical and Surgical Sciences, Data Analytics Research Center, University Magna Græcia, 88100 Catanzaro, Italy; 2Department of Law, Economics and Social Sciences, University Magna Græcia of Catanzaro, 88100 Catanzaro, Italy

**Keywords:** COVID-19, CCTV methodology, network analysis, community detection

## Abstract

Italy was one of the European countries most afflicted by the COVID-19 pandemic. From 2020 to 2022, Italy adopted strong containment measures against the COVID-19 epidemic and then started an important vaccination campaign. Here, we extended previous work by applying the COVID-19 Community Temporal Visualizer (CCTV) methodology to Italian COVID-19 data related to 2020, 2021, and five months of 2022. The aim of this work was to evaluate how Italy reacted to the pandemic in the first two waves of COVID-19, in which only containment measures such as the lockdown had been adopted, in the months following the start of the vaccination campaign, the months with the mildest weather, and the months affected by the new COVID-19 variants. This assessment was conducted by observing the behavior of single regions. CCTV methodology allows us to map the similarities in the behavior of Italian regions on a graph and use a community detection algorithm to visualize and analyze the spatio-temporal evolution of data. The results depict that the communities formed by Italian regions change with respect to the ten data measures and time.

## 1. Introduction

COVID-19 has represented the most important modern challenge for the healthcare system. To fight against this global pandemic, different containment measures were implemented, such as lockdowns, closures of borders by many countries, cancellations of sporting and cultural events, and pharmaceutical measures given by vaccines [1]. Furthermore, statistics were declared daily by each country, and databases have been developed to store this data. In Europe, Italy was the country most affected by the epidemic in 2020, with high numbers of COVID-19-related infections and deaths. Furthermore, in 2021, Italy had a high share of people fully vaccinated against COVID-19, with 90% of the population aged over 12 years vaccinated in January 2022. The data on COVID-19 were released daily by the Italian Civil Protection, including spatial information, such as the geographical regions where data are recorded, and temporal information, i.e., the day of measurement.

In a previous work [2], we presented the COVID-19 Community Temporal Visualizer (CCTV), a methodology for the network-based analysis and visualization of COVID-19 data. In detail, the CCTV methodology comprises four steps: (i) The application of statistical tests to identify the regions that present similar/dissimilar behavior with respect to COVID-19 measures; (ii) the building of similarity matrices; (iii) the mapping of each matrix of similarity into a network where each node is an Italian region and each edge depicts similarity connections; (iv) the identification of communities by applying community detection algorithms.

In this work, we applied CCTV methodology to evaluate the impact of the clinical evolution of the COVID-19 pandemic by integrating several types of clinical data with geographical and temporal data by evaluating the evolution of community coherence in relation to different data in the periods 24 February–26 April 2020 and 28 September–29 November 2020.

Afterward, in [3], we implemented a parallel version of CCTV, called Parallel Network Analysis and Communities Detection (PANC), which we applied to analyze the impact of the evolution of the COVID-19 pandemic by integrating clinical data with geographical data in the period 24 February 2020–28 February 2021.

In this work, we wanted to examine the evolution of COVID-19 in Italy in 2020, 2021 and five months of 2022. In particular, we performed a comparative analysis by focusing on five significant periods in which the presence of non-pharmaceutical or pharmaceutical measures of control have alternated, i.e., the first COVID-19 wave (February–May 2020, which, for convenience, we called the first period), the second COVID-19 wave (October 2020–January 2021, which, for convenience, we called the second period), the months following the start of the vaccination (February–May 2021, which, for convenience, we called the *third period*), the warmest months (June–October, 2021, which, for convenience, we called the *fourth period*), and finally, the months in which the infections started to increase again (November–May 2022, which, for convenience, we called the *fifth period*). The interest of this work is to assess the changes and effects of the COVID-19 spread by analyzing the periods with strong control measures and without vaccinations and the periods marked by a vaccination campaign and containment measures. In particular, the goal of this examination is to evaluate the impact of COVID-19 by taking into account the number of COVID-19 patients in hospital, the number of COVID-19 patients in intensive care units, the daily number of subjects in quarantine at home, the number of COVID-19-positive subjects, the number of subjects healed or discharged from hospital, the daily number of deaths, and the daily number of test swabs carried out in Italy. The results showed that the Italian regions responded differently to the evolution of the COVID-19 epidemic in the different observation periods. In fact, the communities extracted are different in the different periods marked by the first and second COVID-19 waves, in the periods in which the containment measures have been adopted and in the periods in which the containment measures have been added to the vaccination campaign. The methodology and its implementation as an R function are publicly available at https://github.com/mmilano87/analyzeC19D (accessed on 1 August 2022).

The rest of the paper is organized as follows: Section 2 discusses the background of community detection in networks and the background of the evolution of the COVID-19 epidemic in Italy. Section 3 presents the CCTV methodology and the application to Italian COVID-19 data. Section 4 presents and discusses the results. Finally, Section 5 concludes the paper.

## 2. Background

In this section, we present the background on community detection and the algorithms for community discovery presented in the literature. Then, we present the background on the evolution of COVID-19 in Italy in 2020–2022.

### 2.1. Background on Community Discovery

Community detection is one of the most popular research areas in a variety of complex systems, such as biology, sociology, medicine, transportation systems, and the internet. The reason is that the community structures, defined as groups of nodes that are more densely connected than the rest of the network, represent significant characteristics for understanding the functionalities and organizations of complex systems modeled as a network [4]. In fact, it is expected that the communities play significant roles in the structure–function relationship. For example, in biological networks such as Protein–Protein Interaction (PPI) networks, the communities represent proteins involved in similar functions; in neuroscience, the communities detected in brain networks indicate regions of interest (ROI) that are active during tasks; in social networks, communities can be groups of friends or colleagues; in the world wide web, communities represent web pages sharing the same topic [5].

Thus, the discovery of communities in these systems has become an interesting approach to figuring out how network structure relates to system behaviors.

Furthermore, since in real-world systems, the complex system evolves over time, the community structure is also affected by temporal modifications. In fact, the evolution of community structure may relate to the growth of the structure when new members join a group. On the other hand, a community structure can be reduced when several members leave a group or because it can be split into two or more groups. Furthermore, a new community can be formed from the merge of different groups, or it can be created from groups of nodes initially disconnected. Finally, a community can end its existence because all the nodes are disconnected, or it can maintain its structure during the temporal evolution [6,7,8].

Over the years, researchers have proposed different algorithms for the detection of communities.

Here, we reported the most used community detection algorithms in the literature. For example, Fast Greedy [9] and Louvain [10] are two algorithms based on modularity optimization for the identification of communities that differ in the way they optimize the modularity score, where modularity measures how well a network decomposes into modular communities.

Fast Greedy [9] starts by setting each node as a single community, and it joins pairs of communities by applying a basic greedy approach. At each step, the communities are added according to the increase in modularity [11]. Fast Greedy joins pairs of communities until the merging of community pairs does not increase the modularity score.

Louvain [10] includes a community aggregation step to improve the community detection process. The algorithm joins a node with each one of its neighboring communities according to the increase in modularity; otherwise, the node remains in its initial community. Louvain terminates the procedure when no improvement in modularity is obtained. After that, a new network is built whose nodes represent the detected communities and inter and intra-community edges are represented by weighted edges and self-loops, respectively.

The multilevel algorithm [10] uses a greedy approach for optimizing modularity. At first, a community is assigned to each node. Then, according to the increase in modularity, the multilevel algorithm inserts a node into a community formed by its neighbors.

WalkTrap [12] is a hierarchical clustering algorithm that applies a random walk distance measure. Initially, WalkTrap computes the distances between all adjacent nodes in the network. Then, it starts with a node and randomly selects one of its neighbors; it merges them in a community, and it updates the distances between communities. The assumption is that short random walks remain in an equal community.

InfoMod [13] and InfoMap [14] represent the community as parts showing regularities in topology. A community is considered better when it has high compactness and low information loss.

The InfoMod and InfoMap algorithms represent the community structure in different ways. InfoMod represents the community structure as community matrices and membership vectors associating each node to a community, whereas InfoMap depicts the communities by considering two levels: the first one to categorize a community within the network, and the second one to categorize nodes within the communities.

The edge betweenness algorithm [15] is based on edge betweenness, which is a generalization of Freeman’s betweenness centrality measure [16]. The assumption is that the edges that connect communities show a high value of edge betweenness measure. Thus, by deleting the edges with high edge betweenness, the community topology is detected. The Spinglass [17] algorithm is based on physical spin glass models (i.e., the model that describes a magnetic state characterized by randomness, besides cooperative behavior in freezing of spins at a temperature). The algorithm aims to discover the ground state of a spin glass model on the basis that the edges should link nodes with an equal spin state, i.e., equal community. Otherwise, the nodes with different spin states should be disconnected.

The label propagation algorithm [18] detects the communities, considering how the information is transmitted in the network. The assumption is that the nodes of the same community are characterized by high efficiency in the exchange of information. The algorithm initializes a diverse label, i.e., community, for each node. Afterward, it randomly lists the nodes according to a consecutive order. Then, by considering the list, the node is labeled as most of its neighbors. The process ends when all neighbors nodes have the same label. An extended version of the label propagation algorithm is the Community Overlap Propagation Algorithm (COPRA) [19], which enables the detection of communities in weighted and bipartite networks in addition to unweighted ones.

MarkovCluster algorithm [20] works by simulating a stochastic (Markov) flow in a weighted network, where the nodes are data points while the adjacency matrix stores the edge weights. When the algorithm converges, it produces the new edge weights that define the new connected components of the graph (i.e., the clusters).

The leading eigenvector algorithm [21] computes the eigenvectors of the modularity matrix for the optimization of the modularity score. The algorithm computes the leading eigenvector and then splits the network in order to maximize the modularity based on the leading eigenvector.

The algorithms discussed so far are usually applied to static networks for community detection. In literature, there exist different methods for community detection with respect to time. These approaches can be summarized in three categories. The first one regards *traditional static community detection* [6,22] methods. In this method, the network evolution is divided into different timeframes, and the extraction of communities is obtained by traditional static community detection methods. The second class is the *evolutionary clustering method* [23], which aims to find the best community topology that depicts the network at a specific time and to evaluate the similarity of a current community with the structure of a previous time by adding a cost related to temporal smoothness. Finally, *the incremental clustering method* [24] applies the community topology of the first temporal interval to conform to the community property of incremental nodes for the rest of the temporal intervals.

### 2.2. COVID-19 Spread in Italy

At the outbreak of the pandemic, Italy was one of the countries in the world to have suffered the impact of COVID-19 on the number of deaths and the number of infected. To face the epidemic crisis in 2020, the Italian government has adopted non-pharmaceutical measures of control and strong containment policies, such as full lockdown; then in 2021, Italy supported the national COVID-19 prevention planning based on the execution of the vaccination campaign [25]. The impact of vaccination has been noticeable in preventing infections, hospitalizations and deaths. From February 2020 to May 2022 there are five peaks of coronavirus infections, five periods in which the number of new positives has risen and then thanks to the lockdown restrictions, or to the warm season and/or to anti-COVID-19 vaccines it has dropped. An emerging picture shows that vaccines have lowered the number of hospitalized and deceased but the spread of the COVID-19 variants has contributed to a new increase in the number of positive subjects and in the number of hospitalized.

Analyzing the history of the pandemic in Italy from 2020 to May 2022, it is possible to identify 5 significant periods.

*The first period* begins in February 2020 and ends in May 2020 that corresponds to the first wave of COVID-19. Over these months, the cases rise by reaching the peak of infections (#6557 cases) on 21 March 2020. The hospitals fill up with peak admissions (#29,010) on 4 April 2020 and intensive care peak (#4068) on 3 April 2020. On 27 March 2020, the peak of deaths (#969) is reached, whereas the processed swabs are still very few (#73,254 on 11 March 2020). From May to autumn, the effect of the first lockdown, the warm season and the lack of more contagious variants make Italy live relatively peaceful months.

*The second period* begins in October 2020 and ends in January 2021 that corresponds to the second wave of COVID-19. This second period lasts until the start of the vaccination campaign. Over these months, the peak of infections is reached 13 November 2020 with 40,896 positive subjects The hospitals fill up with peak admissions (#29,010) on 23 November 2020 and intensive care peak (#3848) on 25 November 2020. On 3 December 2020, the peak of deaths (#994)is reached and On 30 January 2021, the number of carried out was 298,010. The beginning of the anti-COVID-19 vaccination campaign marks a moment of great hope but at the same time the first stages of vaccinations in Italy are slow and infections and deaths continue to remain at very high numbers.

*The third period* begins in February and in May 2021. This period leads to the warm season and the first re-openings at the end of April after a winter practically in lockdown. The number of positives decreases and, at the same time, the processed swabs increase. Total hospitalizations also decreased in this third period. This period is characterized by a: infection peak (#26,793) recorded on 2 March 2021, a peak admissions (#29,337) on 6 April 2021, an intensive care peak (#572) on 6 April 2021, a peak of deaths (#630) on 7 April 2021 and a peak swabs of #378,463 on 5 March 2021. The summer season arrives, the infections between June and July almost disappear, but then the variants arise and the infections start to rise again. However, the results of the vaccination campaign are beginning to be seen.

*The fourth period* begins in June and in May October 2021. Over these months, the variants are worrying but the warm season and vaccination coverage seem to have an effect. The restrictions are loosening a lot. This period is characterized by a: peak of infections (#7824) recorded on 27 August 2021, a peak in admissions (#430) on 7 September 2021, an a intensive care peak (#572) on 5 September 2021, a peak (#75) deaths on 31 August 2021 and a peak swabs of #357,491 on 5 September 2021.

*The fifth period* begins in November 2021 to May 2022. Over these months, especially from mid-November, the cases begin to rise considerably, but the situation in hospitals, a real parameter to be taken into consideration, seems to be under control. The arrival of the COVID-19 variant brings the new infections to unprecedented peaks. The vaccination obligation is triggered for different categories of workers, the swabs processed increase, reaching almost one million per day.

This period is characterized by a: peak of infections (#50,599) recorded on 24 December 2021, a peak in admissions (#8812) on 24 December 2021, an intensive care peak (#1038) on 24 December 2021, a peak (#168) deaths on 24 December 2021and a peak swabs of 929,775 on 24 December 2021.

## 3. CCTV Methodology

CCTV (COVID-19 Community Temporal Visualizer) methodology ensures the network-based analysis and visualization of generic input dataset organized as collection of homogeneous measures (e.g., COVID-19 data in similar regions).

CCTV methodology design is general and for this reasons it can be applied for the analysis disparate types of data. We implement our methodology using R software [26]. The CCTV methodology as R function is available at https://github.com/mmilano87/analyzeC19D (accessed on 1 August 2022). CCTV function requires the upload igraph libraries [27].

CCTV methodology comprises four steps:1.Building of the similarity matrix: CCTV function takes as input the datasets consisting of the collected data for all regions. Then, the user can select: (i) the kind of aggregate data on which analysis will perform, i.e., Hospitalised with Symptom data, and (ii) the observation period on which he/she want to focus the analysis. After that, CCTV function applies the Wilcoxon test to compute the pair-wise similarity among the regions regarding the selected kind data, i.e., Hospitalised with Symptom data related to Abruzzo vs. Hospitalised with Symptom data related to Basilicata. Then, CCTV builds a similarity matrix that records the *p-value* resulting of the Wilcoxon test for each (i,j) region. At the end of this step, CCTV enable to save as output the built similarity matrix in a text format.2.Mapping similarity matrices to networks-Network building: Starting from the similarity matrix, the CCTV function builds a network related the selected data.3.Network analysis over time: CCTV function builds the networks related to the selected aggregate data in different time intervals. At the end of this step, CCTV function plots the built network. At the end of this step, CCTV enable to save as output the network in image format.4.Community detection: CCTV applied a community detection algorithm to mine the communities on the network built in the previous step. At the end of this step, CCTV function plots the detected communities and it enables to save as output the comunities in image format.

Figure 1 shows the pipeline of methodology and Algorithm 1 shows these steps.

**Algorithm 1:** CCTV Methodology Pseudocode
**Data**: *D* //( Dataset (
**Data**: *w* //( Similarity Measure (
**Result**: *C* (Network Communities )
M←SimilarityMatrix (*D*,*w*);
N←SimilarityNetwork (*M*);
C←CommunityDetection (*N*)
Return (C);


In the following, we treat in detail the CCTV steps by showing the application of our methodology to analyze the Italian COVID-19 dataset.

### 3.1. Dataset

We applied CCTV methodology on the COVID-19 dataset released daily by the Italian Civil Protection at https://github.com/pcm-dpc/COVID-19 (accessed on 16 May 2022). The overall database contains a dataset for each day, starting from 24 February 2020 and for each Italian region. The Italin regions are: Abruzzo, Basilicata, Calabria, Campania, Emilia, Friuli, Lazio, Liguria, Lombardy, Marche, Molise, Piedmont, Puglia, Sardinia, Sicily, Toscana, Umbria, Valle d’Aosta, Veneto, plus the autonomous provinces of Bolzano and Trento, for a total of 21 regions. The dataset of Regions in a time point (day) contains the following 10 data measurements:Hospitalised with Symptoms, as regards the daily the number of COVID-19 patients in the hospital;Intensive Care, as regards the daily number of COVID-19 patients in Intensive Care Units;Total Hospitalised, as regards the daily sum of Hospitalised with Symptoms and Intensive Care measured;Home Isolation, as regards the daily number of subjects in quarantine at home;Total Currently Positive, relating to the daily number of COVID-19 positive subjects;New Currently Positive, as regards the daily number of COVID-19 positive subjects;Discharged/Healed, relating to the daily number healed or discharged from hospital subjects;Deceased, as regards the daily number of deaths;Total Cases, as regards the daily number of subjects affected by COVID-19;Swabs, as regards the daily number of test swab carried on COVID-19 positive subjects and on suspected COVID-19 positivity.

For each time point, the dataset of a region is a vector containing 10 integer values, e.g., at time t=1, $D_{1}^{1}$ = $D_{Lombardia}^{1}$=[x,y,…]. The data occupies 2 Gbytes of memory.

For all analysis we focus on five periods: February–26 May 2020 (that for convenience we called *first period*), October 2020–January 2021 (that for convenience we called *second period*), February–May 2021 (that for convenience we called *third period*), June–October 2021 (that for convenience we called *fourth period*), November–May 2022 (that for convenience we called *fifth period*), and then we compared these periods.

We decided to focus the analyzes on these five observation periods because in Italy from February 2020 until May 2022 there are five peaks of coronavirus infections in which the number of new positives has risen and then thanks to the lockdown restrictions, or to the warm season and/or to anti COVID-19 vaccines it is dropped.

### 3.2. Building of Similarity Matrices

The first step of CCTV methodology consists of the build of similarity matrix. Before this, the needs to identify of similarity measure arises. First of all, we decided to use a non-parametric test for the similarity matrix building, after that we have evaluated the distribution of each type of data by applying Pearson’s chi-square test (*p*-value resulted less than 0.05).

In particular, we chose and applied of the Wilcoxon Sum Rank test [28] test to compare the Italian regions with the aim to evidence statistically similar distributions among them. The Wilcoxon test enables to assess the difference among conditions when the samples are correlated.

Once similarity measure (Wilcoxon test) is applied to dataset *D*, a similarity matrix *M* is built. Let’s matrix *M* for dataset *D* over a time period T, each (i,j) element represents a value obtained by performing a similarity measure. A typical in input dataset $D_{1}\ldots D_{n}$ is a collection of measured data varying a long time. Usually, a generic $D_{i}$ refers to data collected in location *i* (i.e., region or geographical position).

So, the dataset *i* means the dataset collected at Region *i*.

So, we applied a statistical test, the Wilcoxon test, to input dataset $D_{i}$ in order to build a similarity matrix. In detail, the (i,j) value of the similarity matrix, related to data $D_{i}$ (for example deceased data), describes the *p*-value of the Wilcoxon test achieved by applying the test on a given measure (e.g., the number of deceased) of the region *i* versus the region *j* in a given time interval. In this work, we considered the *p*-value as a measure of similarity, i.e., we considered the conventional significance threshold of 0.05 with the goal to build similarity matrices reporting the statistical significance among the COVID-19 data for each region.

So, a lower *p*-value implies that two regions are different according to that measure. Otherwise, a higher *p*-value implies that regions show a similarity according to that measure. We considered the conventional significance threshold of 0.05, and for this reason, we built similarity matrices that contain only *p*-values ≥ 0.05. We mapped the *p*-values < 0.05 equal to zero. An example of the similarity matrix definition is reported in Figure 2.

Thus, we constructed ten similarity matrices for all COVID-19 data and for each time interval, that report the statistical comparison among a couple of regions.

Table 1 reports an example of similarity matrix related to Deceased data in the second period (October 2020–January 2021). All similarity tables related to Italian COVID-19 data are computed for all five observation periods are reported in Appendix A, for the lack of space.

### 3.3. Converting Similarity Matrix to Network

The second step consists of the mapping the similarity matrix into network, i.e., the building of similarity network. So, each similarity matrix M(i,j) is mapped to a network *N*, whose nodes are the Italian regions and the edges connect them when the similarity value among two regions (i,j), resulting in the matrix, exceeds the similarity threshold, i.e., *p*-value > 0.05 The weights of the edges result inversely proportional to similarity values. Thus, we mapped each similarity matrix M(i,j) into a network N [29]. The nodes are the Italian regions, and the edges link two regions (i,j) when the *p*-value is greater than the threshold, otherwise (*p*-value < 0.05) no edge is added. Each edge is weighted according to *p*-value resulting from the Wilcoxon test. In this way, the edge length in the network corresponds to weight and it results inversely proportional to similarity.

### 3.4. Network Analysis over Time

The third step consists of the building of the network at different observation period. Starting from the consideration that the COVID-19 data present a temporal evolution, for each one, the corresponding networks at diverse observation period are built.

We performed a temporal analysis by building ten networks related to the data measures (Hospitalised with Symptoms, Intensive Care data, Total Hospitalised, Home Isolation, Total Currently Positive, New Currently Positive, Discharged/Healed, Deceased, Total Cases, Swab) by considering each observation period discussed above. The data for each considered period resulting by the aggregation of single day, for example the sum of day number of Hospitalised with Symptoms from 24 February to 26 May 2020.

### 3.5. Community Detection and Temporal Evolution

In the last step the extraction of the communities on the built networks is performed. In this step, an algorithm for community detection is applied to identify communities, i.e., groups of regions sharing similarity, on the networks built and for each observation period. The use of an appropriate community detection algorithm is important to achieve the best results. For this aim, we used the Walktrap community finding algorithm [12]. Walktrap is able to identify subgraphs with high density, i.e., communities, in a network through random walks. We selected the Walktrap community detection algorithm because it outperforms other methods as discussed in [30]. Thus, for each network related to different time intervals, we identify of regions that construct a community according to their similarity.

Figure 3, Figure 4, Figure 5, Figure 6, Figure 7, Figure 8, Figure 9, Figure 10, Figure 11 and Figure 12 show the identified communities related to Italian COVID-19 data for all observation periods.

## 4. Results and Discussion

In this section, we analyze the temporal evolution of the detected communities over the five observation periods with the goal to highlight if the communities may be diverse according to different data analyzed and to different observation period when considering the same data. A central question in COVID-19 outbreak is the analysis of the dynamic of COVID-19 evolution and the comparison of the significant of containment measures non-pharmaceutical such as, lockdowns and/or pharmaceutical measures i.e., vaccines in Italy. So, our aim is assessing the effects of lockdown and COVID-19 vaccines on the Italian regions by evaluating the evolution of the communities and the similarity of dissimilarity of the community according to COVID-19 measure.

### 4.1. Trend of Hospitalised with Symptoms Network Communities

We analyze Figure 3 that shows the development of Hospitalised with Symptoms Network Communities. In the first period, 6 communities are identified (Figure 3a): (1) Lazio and Veneto, (2) Trento, Friuli, Bolzano, (3) Campania and Puglia, (4) Calabria, Sardegna, Umbria, Valle d’Aosta, (5) Emilia and Piemonte, (6) Marche, Toscana and Liguria and 6 communtiy formed by a single region (b): (1) Lombardia, (2) Abruzzo, (3) Molise, (4) Basilicata, (5) Sicilia.

In the second period, the regions leave the previous communities, and they move to other ones. For example, 5 communities are identified (Figure 3b): (1) Puglia, Toscana and Sicilia, (2) Trento, Calabria, Bolzano, Umbria (3) Campania and Veneto, (4) Basilica and Valle d’Aosta, (5) Sardegna, Marche, Friuli and Abruzzo, and 7 communtiy formed by a single region: (1) Lombardia, (2) Emilia, (3) Molise, (4) Lazio, (5) Piemonte, (6) Liguria. In Figure 3c all identified communities after three weeks are reported: (1) Friuli, Abruzzo, (2) Emilia, Lazio and Piemonte, (3) Calabria, Umbria and Sardegna, (4) Toscana and Sicilia, (5) Liguria and Marche, (6) Molise, (7), Veneto, (8) Valle d’Aosta, (9) Lombardia, (10) Bolzano, Trento, Basilicata.

In the fourth period, 10 communities are extracted as Figure 3d depicts. (1) Campania and Emilia, (2) Umbria and Basilicata, (3)Piemonte and Puglia, (4) Lombardia, Sicilia, Lazio, (5) Bolzano, Trento and Friuli,(6) Marche, Abruzzo, Liguria, (7) Toscana, (8) Molise, (9) Valle d’Aosta, (10) Calabria, Sardegna and Veneto.

Figure 3e shows the development of the communities in the fifth period. (1) Veneto, (2) Molise, (3) Marche, (4) Valle d’ Aosta, (5) Toscana, (6) Calabria, Friuli, Sardegna, Abruzzo (7) Puglia and Liguria, (8) Lombardia, Lazio, Piemonte, Emilia, (9) Puglia and Liguria, (10) Umbria, Trento, Bolzano, (11) Campania and Toscana. Therefore it is possible to note that: in the first two waves, the Lombardia forms a single community, which is consistent as it was the region most affected by the epidemic; Molise forms a single community in all periods of observation and Emilia and Piedmont exhibit similar behavior in the first, third and fifth periods.

### 4.2. Trend of Intensive Care Network Communities

In Figure 4 the evolution of Intensive Care Network Communities is reported. In the first period (Figure 4a), 8 communties are detected: (1) Calabria, Valle d’Aosta, Molise, Basicilata (2) Piemonte and Emilia, (3) Friuli, Sardegna, Bolzano, umbria (4) Trento, Abruzzo, Sicilia, (5) Campania and Puglia (6) Marche and Liguria, (7) Veneto, Toscana and Lazio (8) Lombardia.

In the second period (Figure 4b), some region leaves the communty to form a single community such as Sicilia, Valle d’ Aosta, Liguria, Molise, Marche and Basilicata. Lombardy continues to represent a single community, whereas other regions move to different communities.

In the third period (Figure 4c), the structure of network becomes sparse. In fact almost all regions that form a single community.

In the fourth period (Figure 4d), some regions form a single new community, whereas other ones joint with previous communities. The extracted communities are: (1) Calabria, (2) Valle d’Aosta, (3) Sicilia, (4) Basilicata, Molise (5) Lazio and Lombardia (6) Puglia, Piemonte, Veneto, Campania (7) Sardegna, Liguria, Marche (8) Umbria, Bolzano, Abruzzo, Friuli, Trento.

The fifth period (Figure 4e) reports 10 mined communities (1) Valle d’Aosta, Basilicata, (2) Lombardia, (3) Molise, (4) Lazio, (5) Lazio, (6) Umbria, (7) Toscana, Piemonte, (8) Campania, Puglia, Marche, (9) Liguria, Friuli, Sardegna, Calabria, Trento, (10) Bolzano, Abruzzo. Therefore, it is possible to infer that in each period the regions formed among themselves different communities.

### 4.3. Trend of Total Hospitalised Network Communities

Figure 5a depicts the development of Total Hospitalised Network Communities in the first period. The first community comprises Basilicata; the second one is represented by Lombardia; the third one is represented by Molise; Abruzzo and Sicilia form the forth community; Trento, Friuli, Bolzano form the fifth community; the sixth one is formed by Marche, Toscana and Liguria; the seventh is composed of Puglia and Campania; the eighth community group Piemonte and Emilia; the ninth one comprises Valle d’Aosta, Umbira, Calabria e Sardegna, tenth community is composed by Veneto and Lazio. In the second period, the detected communities are different respect to the firs ones, as reported in Figure 5b. The first communuty comprises Valle d’Aosta and Basilicata; the second one is represented by Emilia, Veneto, Campagnia; the third one groups Toscana, Puglia, Sicilia; the fourth one comprises Calabria, Bolzano, Trento, Umbria; the fifth is formed by Marche, Abruzzo, Sardegna, Friuli; finally Lombardia, Piemonte, Liguria, Molise and Lazio form single community.

In the third period, the number of communities further grow, as reported in Figure 5c. Basilicata, Lombardia, Molise, Valle d’ Aosta, Bolzano, Trento, Veneto and Puglia form single community. Sicilia and Toscana form a sixth community. The tenth communtity is formed by Liguria and Marche, the eleventh community is composed by Piemonte, Emilia and Lazio; the twelfth community community is formed by Calabria, Sardegna and Umbria. Friuli and Abruzzo form the last community.

In the fouth period, the number of communities further decline, as reported in Figure 5d: (1) Toscana, (2) Valle d’Aosta, (3) Molise (4) Umbria, (5) Basilicata, (6) Veneto, Sardegna, Calabria, (7) Liguria, Abruzzo, Marche, (8) Friuli, Bolzano, Trento, (9) Puglia, Piemonte, (10) Lazio, Lombardi, Sicilia, (11) Emilia, Campania.

The fifth period (Figure 5e) reports 12 mined communities as previous period, but formed by different regions: (1) Valle d’Aosta, (2) Valle d’Aosta, (3) Puglia, (4) Molise, (5) Liguria, (6) Marche, (7) Sicilia, (8) Veneto, Toscana, Campania (9) Friuli, Sardegna, (10) Umbria, Trento, Bolzano, Abruzzo, (11) Piemonte and Emilia, (12) Lazio and Lombardia.

It is possible to notice that in all periods Molise forms a single community, where as, the communities mined in fourth and fifth periods are similar.

### 4.4. Trend of Home Isolation Network Communities

Figure 6a presents the mined communities of the Home Isolation Network. In the first period, 8 communties are detected: (1) Veneto, Emilia, Piemonte, (2) Abruzzo, Trento, Friuli, (3) Lazio and Marche, (4) Toscana, (5) Calabria, Sardegna and Bolzano, (6) Sicilia, Puglia, Campania, Liguria, (7) Umbria, Valle d’Aosta, Molise, Basilicata, (8) Lombardia.

In the second period (Figure 6b), some region leaves the prevous communty to form a single community such as Campania, Valle d’ Aosta and Basilicata. Lombardy continues to represent a single community, whereas other regions move to different communities.

In the third period (Figure 6c), the structure of network becomes sparse. In fact there are different regions that form a single community, such us (1) Marche, (2) Campania, (3) Lombardia, (4) Valle d’Aosta, (5) Molise, (6) Veneto, (7) Sicilia, (8) Bolzano, (9) Trento and also (10) Basilicata, Liguria and Umbria, (11) Lazio, Puglia, Emilia, (12) Abruzzo, Calabria, Friuli.

In the fourth period (Figure 6d), some regions form a single new community, whereas other ones joint with previous communities. The extracted communities are: (1) Trento, (2) Bolzano, (3) Umbria, (4) Molise, (5) Valle d’Aosta, (6) Abruzzo, (7) Piemonte, (8) Marche, (9) Sardegna and Toscana, (10) Lombardia, Sicilia and Campania, (11) Lazio, Veneto, Emila, (12) Puglia and Calabria, (13) Basilicata, Liguria, Friuli.

The fifth period (Figure 6e) reports 8 mined communities (1) Valle d’Aosta, (2) Lombardia and Lazio, (3) Bolzano, (4) Puglia, (5) Liguria and Marche, (6) Basilica and Molise, (7) Toscana, Piemonte, Emilia, Veneto, Campania, Sicilia, (8) Calabria, Sardegna, Abruzzo, Trento, Umbria, Friuli.

### 4.5. Trend of Total Currently Positive Network Communities

Figure 7 shows the development of Total Currently Positive Communities. Figure 7a depicts the detected community in the first period. The first one comprises Lazio and Toscana; the second community is represented by Piemonte and Emilia; the third one is represented by Abruzzo, Bolzano, Friuli, Trento; Umbria, Valle d’Aosta, Molise and Basilicata form the forth community; Lombardia forms the fifth community; the sixth one is formed by Veneto; the seventh is composed of Calabria, Sardegna; the eighth is formed by Sicilia, Camania and Puglia; the ninth is formed by Marche and Liguria. In the second period, the detected communities are different respect to the firs ones, as reported in Figure 7b. The first communuty comprises Marche, Liguria, Fiuli, Bolzano; the second one is represented by Abruzzo and Sardegna; the third one groups Lazio and Veneto; the fourth one comprises Toscana, Puglia, Sicilia; the fifth is formed by Emilia and Piemonte; finally Basilicata, Lombardia, Trento, Campania, Molise, Valle d’Aosta form single communiy.

In the third period (Figure 7c), many regions exhibit different behavior which is mirrored by the fact that they form individual communities. In fact, only 4 communities that are extracted are made up of three regions. A similar trend is also found in the fourth period (Figure 7d). Figure 7e shows the extracted communities in the fifth period: (1) Bolzano, (2) Valle d’Aosta, (3) Puglia, (4) Molise and Basilicata, (5) Lazio and Lombarida (6) Trento, Friuli, Umbria, Sardegna, Calabria, Abruzzo, (7) Toscana, Emilia, Piemonte, Veneto, Sicilia, Campania.

Therefore it is possible to note that: in the first two waves and in the month after vaccination campaign the Lombardia forms a single community, and Emilia and Piedmont exhibit similar behavior as well as Calabria and Bolzano.

### 4.6. Trend of New Currently Positive Network Communities

Figure 8a depicts the development of New Currently Positive Communities in the first period. The first one comprises Basilicata and Molise; the second community is represented by Piemonte and Emilia; the third one is represented by Abruzzo, Sicilia, Friuli, Trento, Puglia, Campania; Ligria, Toscana, Lazio, Marche form the forth community; Lombardia forms the fifth community; the sixth one is formed by Veneto; the seventh is composed of Calabria, Umbria, Sardegna, Valle d’Aosta, Bolzano. In the second period, the detected communities are different respect to the firs ones, as reported in Figure 8b. The first communuty comprises Marche, Abruzzo, Sardegna, Umbria, Bolzano, Calabria; the second one is represented by Emilia, Lazio, Campagnia and Piedmont; the third one groups Toscana, Puglia, Sicilia; the fourth one comprises Valle D’Aosta; the fifth is formed by Liguria and Friuli; finally Basilicata, Lombardia, Trento, Veneto form single community.

In the third period, the number of communities further grow, as reported in Figure 8c. Basilicata, Lombardia, Molise, Lazio, Valle d’ Aosta form single community. Sicilia and Toscana form a sixth community. The seventh communtity is formed by Emilia and Marche, the eighth community is composed by Piemonte, Veneto and Puglia; the ninth community is formed by Trento, Sardegna, Bolzano and Ubria. Calabria, Liguria and Abruzzo form the last community.

In the fouth period, the number of communities further decline, as reported in Figure 8d: (1) Sicilia, (2) Molise and Valle d’Aosta, (3) Marche, Liguria, Abruzzo, Umbria, Friuli (4) Campania, Emilia, Toscana, Lazio, Veneto, Lombardia, (5) Trento, Basilicata, Bolzano and Marche, (6) Piemonte, Puglia, Sardegna, Calabria.

The fifth period (Figure 8e) reports 6 mined communities as previous period, but formed by different regions: (1) Valle d’Aosta, (2) Lombardia and Lazio, (3) Bolzano, (4) Molise and Basilicata, (5) Liguria, Marche, Toscana, sicilia, Puglia, Veneto, Emilia, Campania, Piemonte (6) Trento, Friuli, Umbria, Sardegna, Calabria, Abruzzo.

By comparing the communities extracted in the fifth period on New Currently Positive Network and Total Currently Positive Network is possible to notice a similar trend.

### 4.7. Trend of Discarded or Healed Network Communities

The evolution of Discarded or Healed Network Communities is reported in Figure 9. The communities extracted in first period are reported in Figure 9a. Thus, it is possible to notice that only Lombardy shows a different behavior respect other Italian regions that form community among them. Furthermore, it is possible to notice that over the periods the structure of the network goes from dense to sparse, in this way each region forms a single community (Figure 9b–e).

### 4.8. Trend of Deceased Network Communities

Figure 10 presents the mined communities of the Deceased Network. By analyzing the different periods it is possible to note that apart from some communities formed by few regions in the first and second wave, almost all the regions showed a different behavior forming individual communities.

### 4.9. Trend of Total Cases Network Communities

Figure 11 shows the detected communities Total Cases Network. The first (Figure 11a), second (Figure 11b) and third period (Figure 11c) are characterized by different communities. For example in the first period 6 communities are detected: (1) Abruzzo, Sicilia, Friuli; (2) Marche and Lazio, (3) Calabria and Valle d’Aosta, (4) Emilia and Piemonte (5) Puglia and Trento, (6) Molise and Basilicata and other 8 single communties formed by: Venento, Sardegna, Liguria, Toscana, Lombardia, Umbria, Campania, Bolzano.

In the second period (Figure 11b) the number of communities is reduced because some regions that formed single communities have joined previously formed communities, whereas in the third period (Figure 11d) the previous framework is re-proposed in which different regions form single communities.

Finally, in the fourth (Figure 11d) and fifth period (Figure 11e), all regions form a single community except for (1) Marche and Liguria and (2) Piemonte and Lazio in the fourth period; and (1) Emilia and Piemonte and (2) Campania and Veneto in the fifth period.

### 4.10. Trend of Swab Network Communities

Last Figure 12 reports the evolution of Swab Network Community. As in Total Case Network and Discarded or Healed Network also in Swab Network Community the structure evolves from dense to sparse. So, if in first and second waves Figure 12a,b different regions show a similar behavior according the total number of swabs collected per day, with the introduction of containment measures and vaccines, the Italian regions respond differently to form individual communities in the fourth and fifth periods.

### 4.11. Impact of Containment Measures and Vaccination Campaign

The results evidence the dynamics of the communities varies according to the dynamics of COVID-19. First of all the communities are different both considering the diverse COVID-19 measures and considering the different observation periods. Moreover, the structure of the communities evolves. For example, the communities grow due to joining of regions or the communities reduces due to leaving of regions. This allows to evaluate the changes of community coherence in relation to different data and along different observation period. This aspect is reflected in the community detection analysis in which those regions formed a single community or a community among them. Furthermore, by comparing the communities extracted from the networks in the first observation period and those discovered in the third period, it is possible to notice a substantial diversity among them. In fact the fist period correspond to the fist COVID-19 wave i.e., February–May 2020, whereas the third period i.e., February–May 2021, that are the months marked by containment measures and vaccination campaign. The mined communities are different for each kind of COVID-19 measure and for each two periods and this reflects the different impact that the spread of the virus has had on the Italian regions. In addition, this work also allows to highlight that regions geographically distant may show similar behaviors and form community, for example Calabria and Bolzano in different COVID-19 measures. In conclusion, the rapid spread of COVID-19 has focused attention on the temporal dynamics and effects of the COVID-19 pandemic over time. The CCTV methodology is able to provide a picture of COVID-19 measurements in a given time interval or to represent the temporal evolution both at a regional level and extensible to entire states of a country (e.g., Europe). The results obtained allow to identify the driving factors to control, with adequate health planning, the transmission and impact of the new viral agent and its variants in the environment and society, by analyzing, for example, which events led a region to moving from an initial community, or what events caused a region to move to a more similar community, or what events that caused a region to move to one. For example, these events could be related to pharmaceutical measures, such as vaccinations, that have the potential to keep baseline reproduction numbers low, to relax non-pharmaceutical interventions, and to support the recovery of socioeconomic systems where possible. Therefore, our methodology could be of benefit to design effective health capacity planning strategies to address and/or prevent future waves of COVID-19 and/or similar infectious disease epidemics/pandemics.

### 4.12. Comparison with State-of-Art COVID-19 Research

Over two year, the scientific community focused on the study of virus spread by considering both medical and data Analysis aspects. So, several studies were proposed in the literature. Here, we reported the main works that recurred to network-based representation in order to perform the statistical analysis on COVID-19 data, performed a community detection analysis, and analized COVID-19 pandemic in Italy.

For example, Wang et al. [31] recurred to statistical and network analysis to evaluate an infected cluster of people in different hospitals. The data were represented as heterogeneous network where the nodes represented patients and hospitals and relationships between relatives, friends or colleagues as edges. Network analysis enabled to obtain important information about patients, hospitals and their relationships and to give a guidance for the distribution of epidemic prevention materials. Renardy et al. [32] apply a model based on discrete and stochastic network in a case study of Washtenaw County in Michigan (USA) to forecast the second wave of the COVID-19 pandemic. Kuzdeuov et al. [33] developeds a network-based stochastic epidemic simulator to models the diffusion of a disease through the SEIR states of a population. Kumar [34] used a network-based model to predict the spread of COVID-19, incorporating human mobility through knowledge of migration and air transport. Herrmann et al. [35] modeled as network the human interaction to demonstrate that network topology could improve the predictive power of SIR model of COVID-19 by providing novel insights into the potential strategies and policies for mitigating and suppressing the spread of the virus.

Reich et al. [36] modeled the COVID-19 spread by using a SEIR agent-based model on a graph, by taking into account the following COVID-19 attributes: super-spreaders, realistic epidemiological parameters of the disease, testing, and quarantine policies. Chaudhary et al. [37] analysed the trend, countries affected regionally and the variation of cases at the country level on COVID-19 dataset by applying Principal component analysis on the COVID-19 dataset variables and then, on these ones they applied the unsupervised clustering approach, K-means to find the hidden community structure of countries.

Gibbs et al. [38] apply community detection techniques to human interaction movements network of to identify geographically-explicit movement communities and measure the evolution of these community structures through time. Coccia [39] performed a comparative analysis of the first and second wave of the Coronavirus disease 2019 (COVID-19) to assess the impact on health of people for designing effective policy responses to constrain negative effects of future pandemic waves of COVID-19 and similar infectious diseases in society.

In summary, the work of Chaudhary et al. is the unique work that recur to the community detection method in ored to help in unveiling the patterns of countries and regions where the COVID-19 has impacted in a similar pattern.

CCTV methodology differed from the other previous works because it applied the graph formalism to map the homogeneous data, i.e., COVID-19 data relating to different Italian regions and in different time intervals, into network. In fact, CCTV methodology enables to depict COVID-19 data as networks where each node represents an Italian region and each edge connects statistically similar regions. Furthermore, CCTV methodology enables to conduct community detection task to extract clusters of regions with similar behavior along time. To the best of our knowledge, our work is the first study that provides a network-based representation and visualization of COVID-19 data at the regional level and applies network-based analysis to discover communities of regions that show similar behavior. This makes it difficult to compare CCTV results with other works in literature.

## 5. Conclusions and Future Work

In this work, we applied CCTV methodology to analyze the impact of clinical evolution of COVID-19 pandemic form 2020 to May 2022 in Italy. The results has evidenced that the evolution of the epidemic analyzed considering the containment measures and the diffusion of the vaccination campaign has an impact on the behavior of the various Italian regions. So, CCTV is able to extract different communities in different observation periods that reflect the different policies adopted by the Italian government to fight the epidemic in the 2020, 2021 and mid 2022 years. The present methodology is general and can be applied for the network analysis of different data varying over time. The main limitations of this research concern the data analysis typology. In fact, CCTV is able to provide a graph-based representation of the behavior of the Italian regions that could be extended to different zooms, i.e., by considering the behavior of the individual cities of the Italian regions or by moving the analysis to a wider view, i.e., considering the different European states have faced the pandemic. Thus, the methodology analyzes the homogeneous datasets only by applying a statistical test to find similar/dissimilar datasets. Thus, the researcher that want to extract a deep knowledge from the data, i.e., which event did not cause a region to move from an initial community, or which event that caused a region to move to a more supportive community, for example where fewer were recorded cases or which interventions may improve the behavior of critical regions, should perform a post-processing analysis by applying different metrics.

As future work, we plan to extend the implementation of CCTV, by developing a graphical user interface offering a visualization dashboard that may be used by domain experts and decision makers to analyze the impact of containment measures on a geographical scale.

## Figures and Tables

**Figure 1 biotech-11-00033-f001:**
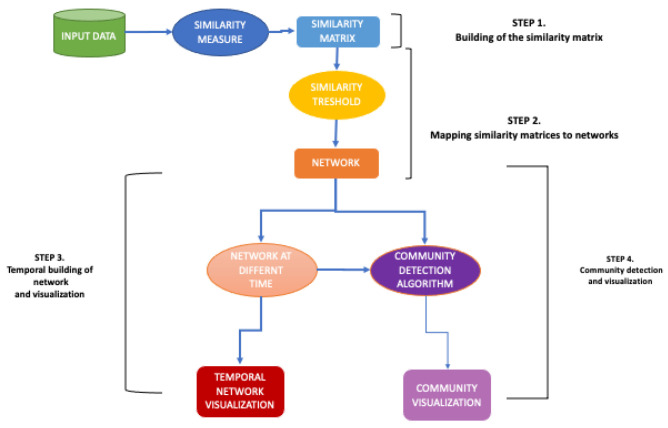
CCTV Methodology pipeline.

**Figure 2 biotech-11-00033-f002:**
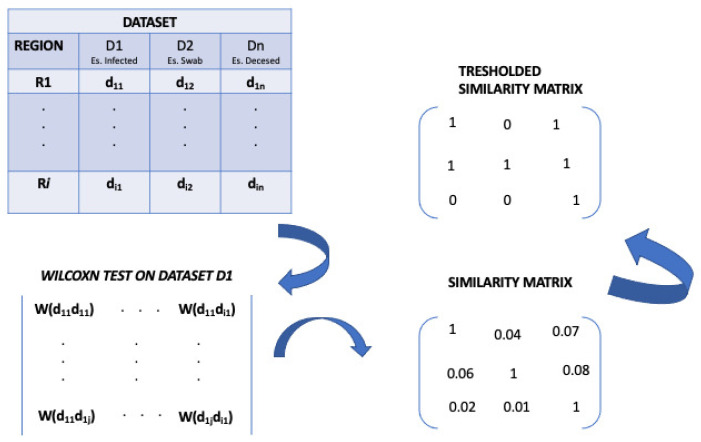
Definition of the similarity matrix.

**Figure 3 biotech-11-00033-f003:**
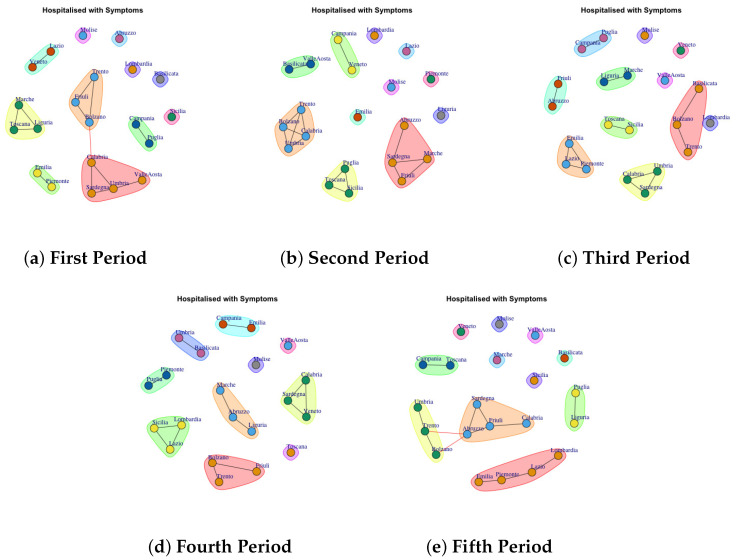
Evolution of Hospitalised with Symptoms Network Communities in the observation periods.

**Figure 4 biotech-11-00033-f004:**
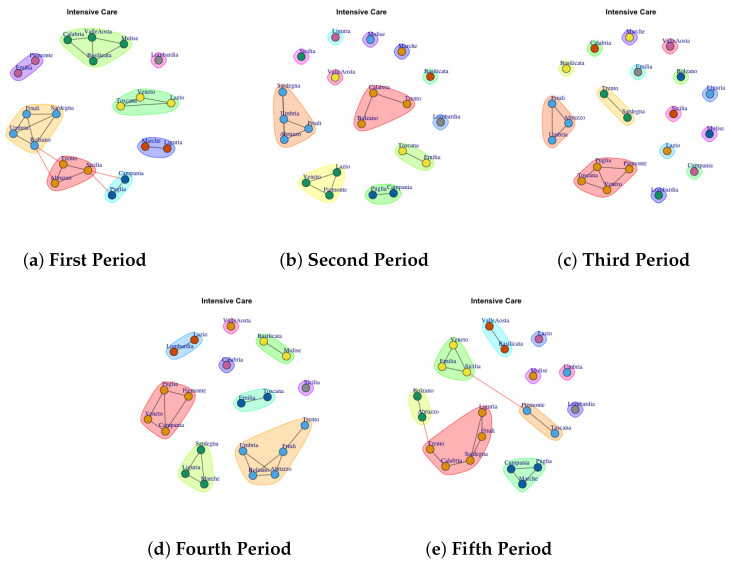
Evolution of Intensive Care Network Communities in the observation periods.

**Figure 5 biotech-11-00033-f005:**
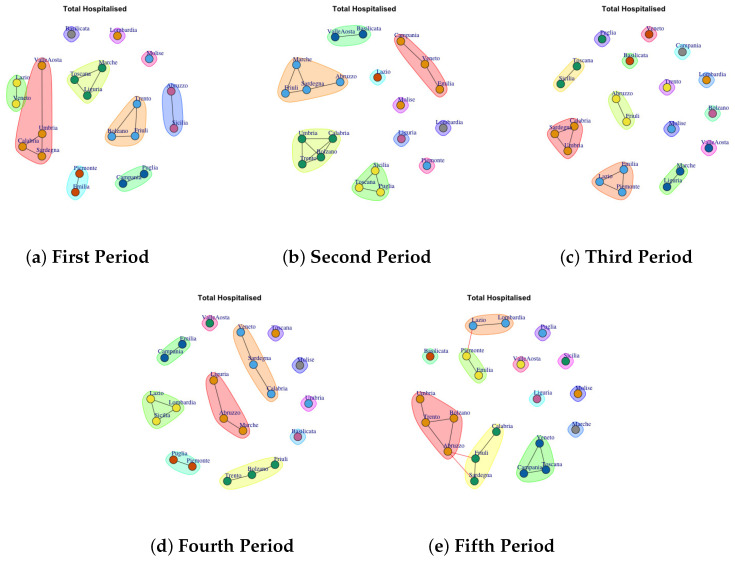
Evolution of Total Hospitalised Network Communities in the observation periods.

**Figure 6 biotech-11-00033-f006:**
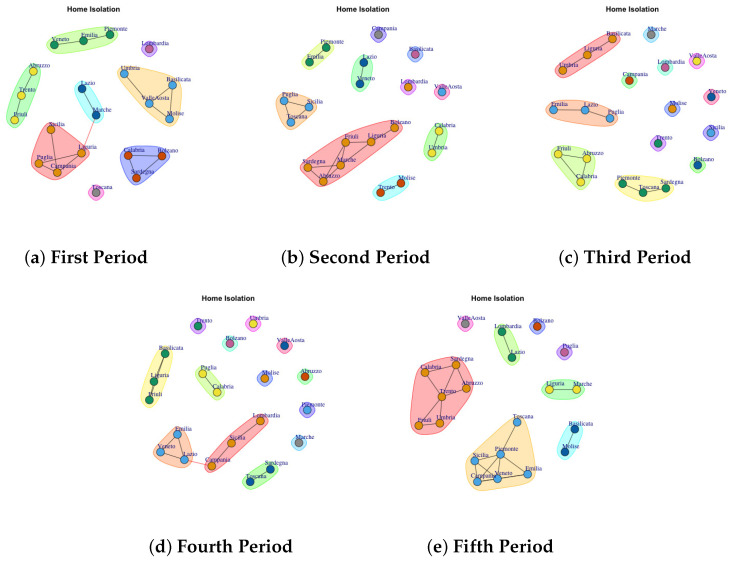
Evolution of Home Isolation Network Communities in the observation periods.

**Figure 7 biotech-11-00033-f007:**
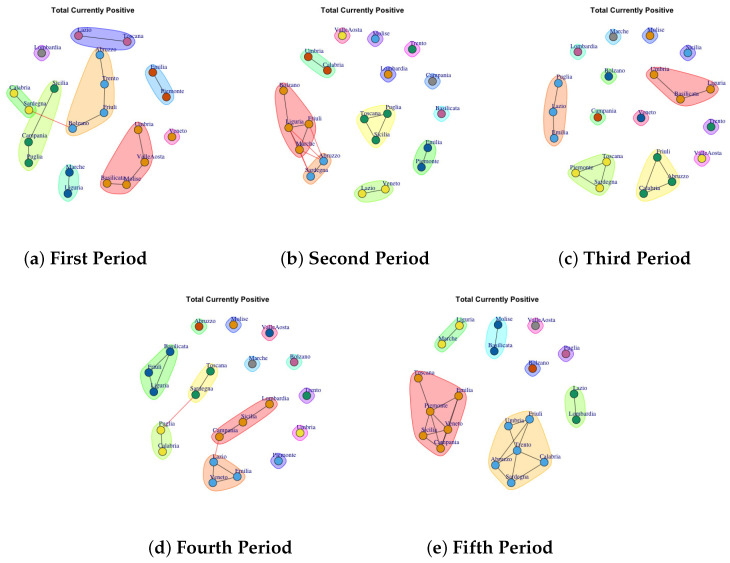
Evolution of Total Currently Positive Network Communities in the observation periods.

**Figure 8 biotech-11-00033-f008:**
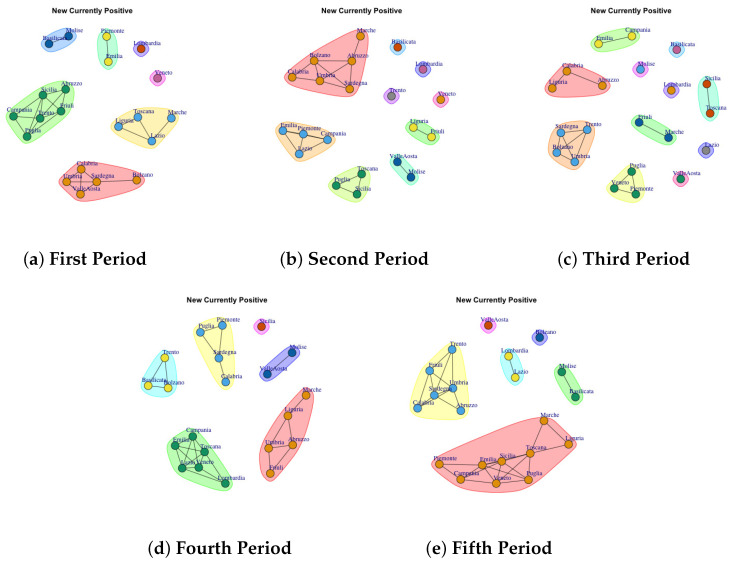
Evolution of New Currently Positive Network Communities in the observation periods.

**Figure 9 biotech-11-00033-f009:**
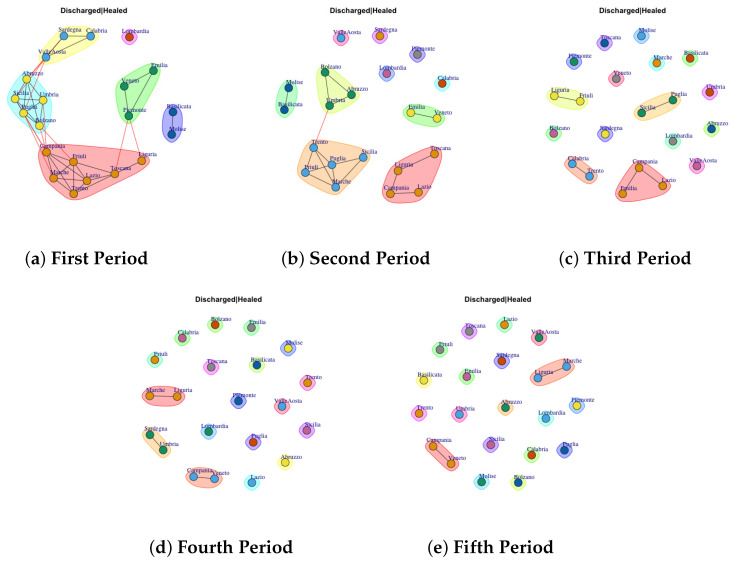
Evolution of Discharged or Healed Network Communities in the observation periods.

**Figure 10 biotech-11-00033-f010:**
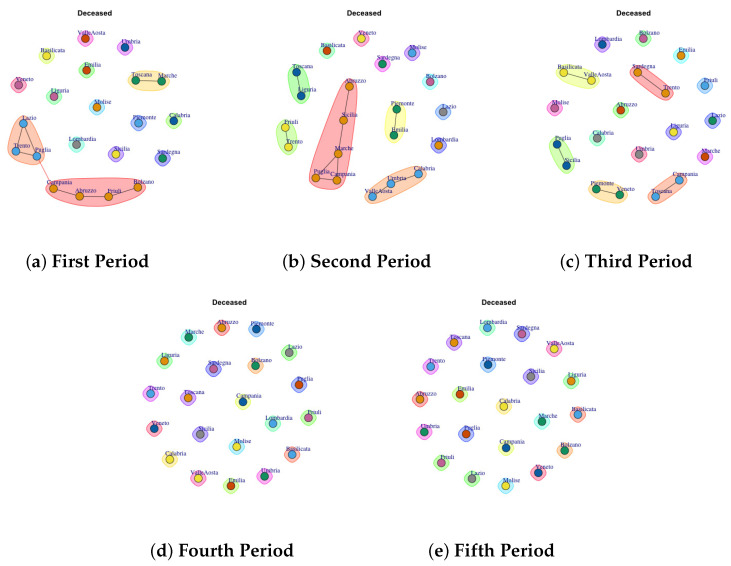
Evolution of Deceased Network Communities in the observation periods.

**Figure 11 biotech-11-00033-f011:**
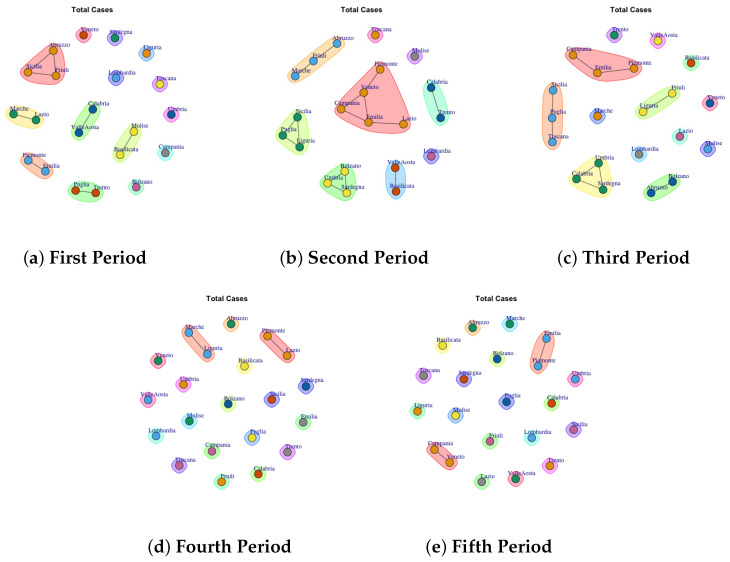
Evolution of Total Cases Communities in the observation periods.

**Figure 12 biotech-11-00033-f012:**
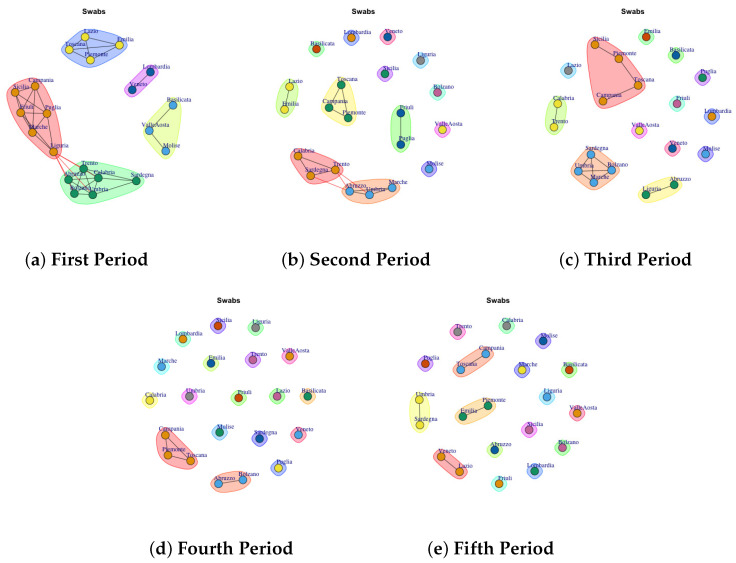
Evolution of Swabs Network Communities in the observation periods.

**Table 1 biotech-11-00033-t001:** Similarity Matrix of Deceased in the second period.

	Abruzzo	Basilicata	Bolzano	Calabria	Campania	Emilia	Friuli	Lazio	Liguria	Lombardia	Marche	Molise	Piemonte	Puglia	Sardegna	Sicilia	Toscana	Trento	Umbria	ValleAosta	Veneto
Abruzzo	1	0	0	0	0	0	0	0	0	0	0	0	0	0	0	0.1	0	0	0	0	0
Basilicata	0	1	0	0	0	0	0	0	0	0	0	0	0	0	0	0	0	0	0	0	0
Bolzano	0	0	1	0	0	0	0	0	0	0	0	0	0	0	0	0	0	0	0	0	0
Calabria	0	0	0	1	0	0	0	0	0	0	0	0	0	0	0	0	0	0	0.15	0	0
Campania	0	0	0	0	1	0	0	0	0	0	0.39	0	0	0.73	0	0	0	0	0	0	0
Emilia	0	0	0	0	0	1	0	0	0	0	0	0	0.32	0	0	0	0	0	0	0	0
Friuli	0	0	0	0	0	0	1	0	0	0	0	0	0	0	0	0	0	0.94	0	0	0
Lazio	0	0	0	0	0	0	0	1	0	0	0	0	0	0	0	0	0	0	0	0	0
Liguria	0	0	0	0	0	0	0	0	1	0	0	0	0	0	0	0	0.27	0	0	0	0
Lombardia	0	0	0	0	0	0	0	0	0	1	0	0	0	0	0	0	0	0	0	0	0
Marche	0	0	0	0	0.39	0	0	0	0	0	1	0	0	0.19	0	0.17	0	0	0	0	0
Molise	0	0	0	0	0	0	0	0	0	0	0	1	0	0	0	0	0	0	0	0	0
Piemonte	0	0	0	0	0	0.32	0	0	0	0	0	0	1	0	0	0	0	0	0	0	0
Puglia	0	0	0	0	0.73	0	0	0	0	0	0.19	0	0	1	0	0	0	0	0	0	0
Sardegna	0	0	0	0	0	0	0	0	0	0	0	0	0	0	1	0	0	0	0	0	0
Sicilia	0.1	0	0	0	0	0	0	0	0	0	0.17	0	0	0	0	1	0	0	0	0	0
Toscana	0	0	0	0	0	0	0	0	0.27	0	0	0	0	0	0	0	1	0	0	0	0
Trento	0	0	0	0	0	0	0.94	0	0	0	0	0	0	0	0	0	0	1	0	0	0
Umbria	0	0	0	0.15	0	0	0	0	0	0	0	0	0	0	0	0	0	0	1	0.6	0
ValleAosta	0	0	0	0	0	0	0	0	0	0	0	0	0	0	0	0	0	0	0.6	1	0
Veneto	0	0	0	0	0	0	0	0	0	0	0	0	0	0	0	0	0	0	0	0	1

## Data Availability

The dataset used in this work is available at https://github.com/mmilano87/analyzeC19D (accessed on 1 August 2022).

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
