# Peer review of "Application of CCTV Methodology to Analyze COVID-19 Evolution in Italy"

_biotech, 2022, doi:10.3390/biotech11030033_

Round 1

Reviewer 1 Report

This study extends a previous work by using CCTV to evaluate how the Italian regions have changed their behavior during some COVID-19 waves with respect to some parameters.

The introduction in very well presented.

The methods, including how the algorithm is applied and how the similarity matrices are built, how they are converted to a network, and how the network is analyzed over time are very well presented.

The result are well presented and discussed.

Author Response

Reviewer #1

This study extends a previous work by using CCTV to evaluate how the Italian regions have changed their behavior during some COVID-19 waves with respect to some parameters.

The introduction in very well presented.

The methods, including how the algorithm is applied and how the similarity matrices are built, how they are converted to a network, and how the network is analyzed over time are very well presented.

The result are well presented and discussed.

Answer: We thanks the reviewer for the positive comment.

Reviewer 2 Report

In this work, the authors applied CCTV methodology to analyze the impact of clinical evolution of COVID-19 pandemic form 2020 to May 2022 in Italy. 

The work is generally descriptive, based on certain statistical approaches. Since there is a lot of data and results in the paper, I would recommend the authors to specify the analysis, and not to repeat the text. Specific result, and conclusion on it, for each period of analysis. 

My main remarks are only about the preparation of the text and formatting. I can not check the results obtained by the authors, although the authors have presented the methodology and its implementation as R function are publicly available at https://github.com/mmilano87/analyzeC19D. 

Minor comments:

5: related to 2020, 2021, e five months of 2022. - typo "e". 

316: So we con - not a complete sentence. 

Unexpectedly the text mentions Table A37: 338 Table A37 reports an example of similarity matrix related to deceased data in the  

I recommend that the authors move all tables (Tables A1-50) to the Supplemental file. Also, these tables can be merged into one table. Including Table 1. Similarity Matrix of Deceased in the second period.

I am sure the authors have to merge Figures 3-12 somehow, there are too many figures. 

The whole work is statistics and the reader just gets lost in the amount of tables and figures. I think the authors can solve this problem, and make one figure and one summary table. Where there is no data in the tables, authors don't have to use them. Only the significant values and significant changes in the figures, leave it alone.

It is easier for me as a researcher to evaluate the work when all the data and results are covered at once.

There is duplication in the text, which in the abstract is in the conclusion and in the introduction. 

Also, I am not quite sure that this paper is relevant to the aim and scope of BioTech. But I would not mind supporting this work, in case of finalizing the text according to my recommendations. 

Author Response

In this work, the authors applied CCTV methodology to analyze the impact of clinical evolution of COVID-19 pandemic form 2020 to May 2022 in Italy. 

The work is generally descriptive, based on certain statistical approaches. Since there is a lot of data and results in the paper, I would recommend the authors to specify the analysis, and not to repeat the text. Specific result, and conclusion on it, for each period of analysis. 

AnswerWe thanks the reviewer to pointing out this. We rewrote the Results and Discussion section by analyzing the results according to each observation period.

Results and Discussion ->

In this section,  we analyze the temporal evolution of the detected communities over the five observation  periods with the goal to  highlight if the communities may be diverse according to different data analyzed and to different observation period  when considering the same data.

A central question in COVID-19 outbreak is the analysis of the dynamic of COVID-19 evolution and the comparison of the significant  of containment measures  non-pharmaceutical such as, lockdowns and/or pharmaceutical measures i.e. vaccines in Italy.

So, our aim is assessing the effects of lockdown and COVID-19 vaccines  on the Italian regions by evaluating the evolution of the communities and the similarity of dissimilarity of the community according to COVID-19 measure.

\subsection{First observation period}

The first period  (February-May,2020)  corresponds to  the first wave of COVID-19.

By analyzing the extracted  communities  on the different COVID-19 data, it is possible to notice that are different cluster of communities formed by an high number of regions. This means that a consistent number of regions presented a similar evolution in dealing with the first wave of COVID-19. For example, by analyzing the trend of Hospitalised with Symptoms Network Community it is possible to notice that  the Emilia and Piedmont exhibit similar behavior as well as in Total Currently Positive Network Communities Emilia and Piedmont exhibit similar behavior and  Calabria and Bolzano too.  Also, in the first period the progression of the Hospitalised with Symptoms Network Communities is similar to the Total Hospitalised Network Communities, as well as Deceased Network Communities, and Total Cases Network Communities report similar evolution.

Furthermore, for all COVID-19 data, Lombardia forms a single community, which is consistent as it was the region most affected by the epidemic, with the exception of Swab data respect to which Lombardy shows a similar behavior with Veneto by forming a community. Finally, Molise forms a single community in all periods of observation.

\subsection{Second  observation period}

 The second  period (October, 2020- January, 2021)  corresponds to the second wave of COVID-19.

 In this period,  the regions leave the previous communities, and they move to other ones, also it is possible to note that the topology evolves from a sparse to a dense structure and this reflects on the discovered communities.

 This means that impact of COVID-19 is different respect to first period.

 In fact, by analyzing Hospitalised with Symptoms Network Communities, Total Hospitalised Network Communities, Total Currently Positive Network Communities, Discharged/ Healed Network Communities, Deceased Network Communities, Total Cases Network Communities, it is possible to notice that the Italian regions represent single communities that is reflected by the sparse network topology. Especially, in Deceased Network, different region forms a single community. Also in  Intensive Care Network Communities some region leaves the community to form a single community.

 Furthermore, by comparing the communities extracted from the networks in the first observation period and those discovered in the second observation period, it is possible to notice a substantial diversity among them. For example, Home Isolation Network Communities and Total Currently Positive Network Communities are different respect to first period.

Lombardy continues to represent a single community in this period.

 \subsection{Third observation period}

 The third  period (February-May, 2021) corresponds to the warm  season and the first re-openings at the end of April after a winter practically in lockdown.

 By analyzing the trend of Hospitalised with Symptoms Network Community, it is possible to notice that  the Emilia and Piedmont exhibit similar behavior, whereas in Total Hospitalised Network   New Currently Positive Network  the number of  communities further grow.

 In Home Isolation Network Communities and  Intensive Care Network, the structure of the network becomes sparse. In fact there are different regions that form a single community,

 Total Currently Positive Network Communities, many regions exhibit different behavior which is mirrored by the fact that they form individual communities.

  About Total Cases Network Communities, if in the second period,  the number of communities is reduced because some regions that formed single communities have joined previously formed communities, in the third period the previous framework is re-proposed in which different regions form single communities.

Therefore, in Total Currently Positive Network Communities, it is possible to note that: in the first two waves and in the month after vaccination campaign, the Lombardia forms a single community, and  Emilia and Piedmont exhibit similar behavior as well as Calabria and Bolzano.

 \subsection{Fourth  and Fifth observation periods}

The  fourth and  fifth periods correspond June-October 2021 ad November, 2021-May, 2022.

These last two periods show a similar trend for all covid-19 measures

For example, by comparing the communities extracted in the fifth period on  New Currently Positive Network and Total  Currently Positive Network is possible to notice a similar trend.

By analyzing  the evolution of Discarded or Healed Network Communities it is possible to notice that only Lombardy shows a different behavior respect other Italian regions that form community among them.

Furthermore, it is possible to notice that over the periods the  structure of the network goes from dense to sparse, in this way each region forms a single community.

In Total Hospitalised Network Communities, it is possible to notice that in all periods Molise forms a single community, whereas, the communities mined in fourth and fifth periods are similar.

By analyzing the mined communities of the Deceased  Network in different periods it is possible to note that apart from some communities formed by few regions in the first and second wave, almost all the regions showed a different behavior forming individual communities.

Also, Total Case Network and Discarded or Healed Network and in Swab Network Community the structure evolves from dense to sparse. So, if in first and second waves  different regions show a similar behavior  according the total number of swabs collected per day, with the introduction of containment measures and vaccines, the Italian regions respond differently to form individual communities in the fourth and fifth periods.

The results evidence the dynamics of the communities vary according to the dynamics of  COVID-19.

 First of all the communities are different both considering the diverse COVID-19 measures and considering the different observation periods.

 Moreover, the structure of the communities evolves. For example, the communities grow due to joining of regions or the communities reduces due to leaving of regions. This allows to evaluate the changes of community coherence in relation to different data and along different observation period.

This aspect is reflected in the community detection analysis in which those regions formed a single community or a community among them.

 Furthermore, by comparing the communities extracted from the networks in the first observation period and those discovered in the  third period, it is possible to notice a substantial diversity among them.  In fact the fist period correspond to the fist COVID-19 wave i.e. February- May 2020, whereas the third period i.e. February- May 2021, that are the months marked by containment measures and vaccination campaign.

The  mined communities are different for each kind of COVID-19 measure and for each two periods, and  this reflects the different impact that the spread of the virus has had on the Italian regions.

In fact, analyzing the structure of the networks it is possible to highlight that when Italy only applied containment measures such as the lockdown dependent on a national policy, different clusters of communities were found formed by a large number of regions. While the vaccination campaign linked to regional policies has brought about a modification of the structure of the networks that from scattered have become dense, for which all the regions formed single communities exhibiting a different behavior between them.

In addition, this work also allows to highlight that regions geographically distant may show similar behaviors and form community, for example Calabria and Bolzano in different COVID-19  measures.

In conclusion, the rapid spread of COVID-19 has focused attention on the temporal dynamics and effects of the COVID-19 pandemic over time.

The CCTV methodology is able to provide a picture of COVID-19 measurements in a given time interval or to represent the temporal evolution both at a regional level and extensible to entire states of a country (e.g. Europe).

The obtained results  allow  to identify the driving factors to control, with adequate health planning, the transmission and impact of the new viral agent and its variants in the environment and society, by analyzing, for example, which events led a region to moving from an initial community, or what events caused a region to move to a more similar community, or what events that caused a region to move to one. For example, these events could be related to pharmaceutical measures, such as vaccinations, that have the potential to keep baseline reproduction numbers low, to relax non-pharmaceutical interventions, and to support the recovery of socioeconomic systems where possible.

Therefore, our methodology could be of benefit to design effective health capacity planning strategies to address and / or prevent future waves of COVID-19 and / or similar infectious disease epidemics / pandemics.

My main remarks are only about the preparation of the text and formatting. I can not check the results obtained by the authors, although the authors have presented the methodology and its implementation as R function are publicly available at https://github.com/mmilano87/analyzeC19D. 

AnswerWe apologize since we were not able to clarify this point. We on github a result folder that contains all the results (similarity matrices, networks and communities) of our work.

Minor comments:

5: related to 2020, 2021, e five months of 2022. - typo "e". 

Answer: We apologize for the issue. We fixed this.

316: So we con - not a complete sentence. 

Answer: We apologize for the issue. We fixed this.

Unexpectedly the text mentions Table A37: 338 Table A37 reports an example of similarity matrix related to deceased data in the  

Answer: We apologize for the issue. We fixed this.

I recommend that the authors move all tables (Tables A1-50) to the Supplemental file. Also, these tables can be merged into one table. Including Table 1. Similarity Matrix of Deceased in the second period.

Answer: We thanks the reviewer to pointing out this. We moved the all tables in the Supplementary file.

I am sure the authors have to merge Figures 3-12 somehow, there are too many figures. 

Answer: We thanks the reviewer to pointing out this. We merged Figures 3-12 in unique Figure 3 and we moved these ones in the Supplementary file.

The whole work is statistics and the reader just gets lost in the amount of tables and figures. I think the authors can solve this problem, and make one figure and one summary table. Where there is no data in the tables, authors don't have to use them. Only the significant values and significant changes in the figures, leave it alone.

Answer: We thanks the reviewer to pointing out this. We summarize the  Figures 3-12 in unique Figure and we moved the all figures in the Supplementary file. About the tables,  we did not create a summary table because the merging of 50  table (10 table related to COVID-19 data x 5 periods) in single one, would have made it unreadable. So, we move all the tables in the Supplementary file. About the  significant values in table, we apologize since we were not able to clarify this point.  We reported in table the values equal to zero because, according to our methodology design, CCTV put to zero all p-values values <0.05.  Since CCTV constructs the networks starting from the similarity matrices contained in the tables, the values equal to zero indicate that there is no edge between the entries (i.e. the nodes) that contain that value. Therefore, we cannot report only the significant values (i.e. p-values> 0. 05) as entries of the tables.

It is easier for me as a researcher to evaluate the work when all the data and results are covered at once.

There is duplication in the text, which in the abstract is in the conclusion and in the introduction. 

Answer: We thanks the reviewer to pointing out this.  

 We rewrote the Abstract ->

Italy represented  one of European countries most afflicted by the COVID-19 pandemic. Over  2020-2022 years, Italy adopted strong containment measures against the COVID-19 epidemic and then, it started an important vaccination campaign. Here, we extended a previous work by applying COVID-19 Community Temporal Visualizer (CCTV) methodology  to  Italian COVID-19 data  related to 2020, 2021, and five months of 2022. The aim of this work was to evaluate how   Italy reacted to pandemic in the first two waves of COVID-19 in which only containment measures such as the lockdown had been adopted, in the months following the start of the vaccination campaign, the months with the mildest weather, and the months affected by the new COVID-19 variants. This assessing was conducted by observing the behavior of single regions. CCTV methodology enables to mapping such similarity information among  Italian region behavior on a graph and then to use a community detection algorithm to visualize and analyze the spatio-temporal evolution of data. The  results  depict the communities formed by Italian regions change with respect to the ten data measures and time. The methodology and its implementation as R function are publicly available at https://github.com/mmilano87/analyzeC19D.

We rewrote the Introduction ->

COVID-19 has represented the most important modern challenge for health system. To fight against this global pandemic, different  containment measures was implemented, such as lockdown, closure of the borders by many countries, cancellations of sporting and cultural events, but also  pharmaceutical measures, given by vaccines [1]. Furthermore,  statistics were declared every day by the countries and databases have been developed to store this data. In Europe, Italy is the country most affected to epidemic in 2020 with high numbers of COVID-19 related infected individuals and deaths. Furthermore, in 2021, Italy has a high share of people fully vaccinated against COVID-19  with 90\% of the population aged over 12 years in January 2022. The data about COVID-19 was released daily by the Italian Civil Protection,  including spatial information such as the geographical regions, where data are recorded, and temporal information, i.e. the day of measurement. In a previous work [2], we presented COVID-19 Community Temporal Visualizer (CCTV), a methodology for the network-based analysis and visualization of COVID-19 data. In detail, the CCTV methodology comprises four steps: i) Application of statistical test to identify the regions that present similar/dissimilar behavior with respect to COVID-19 measures; ii) Building of similarity matrices; iii) Mapping of each matrix of similarity into a network where each node is an Italian region, and each edge depicts similarity connections; iv) Identification of communities by applying community detection algorithms.  In this work, we applied CCTV methodology  to evaluate the impact of clinical evolution of COVID-19 pandemic by integrating several clinical data on geographical and temporal data  and then climate data by evaluating the evolution of communities coherence in relation to different data on the period February 24-April 26, 2020, and  on the period  September 28-November 29, 2020. After that, in [8], we implemented the parallel version CCTV, called Parallel Network Analysis and Communities Detection (PANC), that we applied to analyze the impact of evolution of COVID-19 pandemic by integrating clinical data on geographical data on the period February 24, 2020-February 28, 2021. In this work, we wanted to examine the evolution of COVID-19 in Italy  in 2020 year, 2021 year and five months of 2022 year. In particular, we performed a comparative analysis by focusing  on five significant periods, in which the presence of non-pharmaceutical or pharmaceutical measures of control have alternated, i.e. the first COVID-19 wave (February-May, 2020, that for convenience we called first period),  the second  COVID-19 wave (October, 2020-January, 2021, that for convenience we called  second period), the months following the start of the vaccination (February-May, 2021, that for convenience we called third period), the most warm months (June-October, 2021, that for convenience we called  fourth period), and finally, the months in which the infections start to increase again (November-May, 2022, that for convenience we called  fifth period). The interest of this work is to assess the changing  and effects of COVID-19 spread   by analyzing the periods with strong control measures and  without vaccinations  and  the  periods marked by a vaccination campaign and also containment measures. In particular, the goal of this examination is the evaluation of the impact of COVID-19 by taking into account the number of COVID-19 patients in the hospital, the  number of COVID-19 patients in intensive care units, the daily number of  subjects in quarantine at home, the  number of COVID-19 positive  subjects,  the  number healed or discharged from hospital subjects, the daily number of deaths, the daily number of  test swab carried in Italy. The results showed that the Italian regions responded differently to the evolution of the COVID-19 epidemic in the different observation periods. In fact, the communities extracted are different in the different periods marked by the first and second COVID-19 wave, in the periods in which the containment measures have been adopted and in the periods in which the containment measures have been added the vaccination campaign. The  rest of the paper is organized as follows: Section 2 discusses the background on community detection on networks and the background on the correlation among climate data and COVID-19 epidemic, Section 3 presents the CCTV methodology and the application on  Italian COVID-19 data, Section 4 presents and discusses the results. Finally, Section 5 concludes the paper.

We rewrote the Conclusion ->

In this work, we applied CCTV methodology to analyze the impact of clinical evolution of COVID-19 pandemic form 2020 to May 2022 in Italy. The results has evidenced  that  the evolution of the epidemic analyzed considering the containment measures and the diffusion of the vaccination campaign has an impact on the behavior of the various Italian regions. So, CCTV is able to extract different communities in different observation periods that reflect the different policies adopted by the Italian government to fight the epidemic in the  2020,2021 and mid 2022 years. The present methodology is general and can be applied for the network analysis of different data varying over time. The main limitations of this research concern the data analysis typology. In fact, CCTV is able to provide a graph-based representation of the behavior of the Italian regions that could be extended to different zooms, i.e. by considering the behavior of the individual cities of the Italian regions or by moving the analysis to a wider view, i.e. considering the different European states have faced the pandemic. Thus,  the methodology  analyzes the homogeneous datasets only by applying a statistical test to find similar/dissimilar datasets. Thus, our methodology is able to describe a phenomena. However  the researcher that want to extract a deep knowledge from the data, i.e.  which event  did not cause a region to move from an initial community, or which event that caused a region to move to a more supportive community, for example where fewer  were recorded cases or which interventions may improve the behavior of critical regions, they should recur  to data mining methods.

 As future work, we plan to extend the implementation of CCTV, by developing a graphical user interface offering a visualization dashboard that may be used by domain experts and decision makers to analyze the impact of containment measures on a geographical scale. Also, we plan to develop an extended version that enables to perform data mining and machine learning analysis.

Also, I am not quite sure that this paper is relevant to the aim and scope of BioTech. But I would not mind supporting this work, in case of finalizing the text according to my recommendations. 

Answer: We thanks the reviewer of the comments that we applied on the new version of paper.

Reviewer 3 Report

The article is highly up-to-date and has significant scientific value. It refers to problems that are not only theoretical, but especially practical. The advantages of the article include:

- Abstract that presents the major aspects of the entire paper, ie the overall purpose of the study and the research problem, the basic design of the study; major findings;

- Intruduction, where it is explained clearly what is novel to previous research;

- a properly conducted literature review;

- Methods, where the method of testing for CCTV procedures is indicated;

- Results of the research with the use of graphical forms of high scientific value are presented in detail;

The following are the areas of the paper that should be corrected by the authors:

- in the "Results and Discussion" section, there are in practice only results. The discussion should consist in comparing own research results with similar results of other authors, published in scientific journals. Therefore, authors should provide a confrontation of the achieved results with previously published papers, and add thier opinion of established differences

- Conclusions should clearly specify research limitations, which in the case of such current research become even a priority.

-spelling mistakes should be elimitated

Author Response

The article is highly up-to-date and has significant scientific value. It refers to problems that are not only theoretical, but especially practical. The advantages of the article include:

- Abstract that presents the major aspects of the entire paper, ie the overall purpose of the study and the research problem, the basic design of the study; major findings;

- Introduction, where it is explained clearly what is novel to previous research;

- a properly conducted literature review;

- Methods, where the method of testing for CCTV procedures is indicated;

- Results of the research with the use of graphical forms of high scientific value are presented in detail;

The following are the areas of the paper that should be corrected by the authors:

- in the "Results and Discussion" section, there are in practice only results. The discussion should consist in comparing own research results with similar results of other authors, published in scientific journals. Therefore, authors should provide a confrontation of the achieved results with previously published papers, and add thier opinion of established differences

Answer: We thanks the reviewer to pointing out this. We insert a Comparison with State-of-Art COVID-19 research ->

Over two year, the scientific community focused on the study of virus spread by considering both medical and  data Analysis aspects.

So, several studies were proposed in the literature. Here, we reported the main works that  recurred to network-based representation in order to perform the statistical analysis  on COVID-19 data,  performed a community detection  analysis, and analyzed COVID-19 pandemic in Italy.

For example, Wang et al.  \cite{wang2020statistical}  recurred to statistical and network analysis  to evaluate an infected cluster of people in different hospitals. The data were represented as  heterogeneous network where the nodes represented  patients and hospitals and relationships between relatives, friends or colleagues as edges. Network analysis enabled to obtain important information about patients, hospitals and their relationships and  to give a guidance for the distribution of epidemic prevention materials.

Renardy et al. \cite{renardy2020predicting} apply a model based on discrete and stochastic network in a case study of Washtenaw County in Michigan (USA) to forecast the second wave of the COVID-19 pandemic.

Kuzdeuov et al.\cite{kuzdeuov2020network} developeds a network-based stochastic epidemic simulator to models the diffusion of a disease through the SEIR states of a population.

Kumar  \cite{kumar2020modeling} used a  a network-based model to predict the spread of COVID-19, incorporating human mobility through knowledge of migration and air transport.

Herrmann et al. \cite{herrmann2020using} modeled as network the human interaction to demonstrate that network topology could improve the predictive power of SIR model of COVID-19 by providing novel insights into the potential strategies and policies for mitigating and suppressing the spread of the virus.

Reich et al. \cite{reich2020modeling} modeled the COVID-19 spread by using a SEIR agent-based model on a graph, by taking into account the following COVID-19 attributes: super-spreaders, realistic epidemiological parameters of the disease, testing, and quarantine policies.

Chaudhary et al. \cite{chaudhary2021community} analysed the trend, countries affected regionally and the variation of cases at the country level on COVID-19 dataset by applying   Principal component analysis on the COVID-19 dataset variables and then, on these ones they applied   the unsupervised clustering approach, K-means to find the hidden community structure of countries.

 Gibbs et al. \ref{gibbs2021detecting} apply community detection techniques to  human interaction movements network of  to identify geographically-explicit movement communities and measure the evolution of these community structures through time.

Coccia \cite{coccia2021impact} performed a comparative analysis of the first and second wave of the Coronavirus disease 2019 (COVID-19) to assess the impact on health of people for designing effective policy responses to constrain negative effects of future pandemic waves of COVID-19 and similar infectious diseases in society.

In summary, the work of Chaudhary et al. is the unique work that recur to the community detection method in order to help in unveiling the patterns of countries and regions where the COVID-19 has impacted in a similar pattern.

CCTV methodology differed from the other previous works because it applied the graph formalism to map the homogeneous data, i.e. COVID-19 data relating to different Italian regions and in different time intervals, into network.

In fact, CCTV methodology enables to depict  COVID-19 data as networks where each node represents an Italian region and each edge connects statistically similar regions.

Furthermore, CCTV methodology enables  to conduct community detection task to extract clusters of regions with similar behavior along time.

To the best of our knowledge, our work is the first study that provides a network-based representation and visualization of COVID-19 data at the regional level and applies network-based analysis to discover communities of regions that show similar behavior.

This makes it difficult to compare CCTV results with other works in literature.

- Conclusions should clearly specify research limitations, which in the case of such current research become even a priority.

Answer: We thanks the reviewer to pointing out this.   We added the research limitation in the Conclusion section ->

The main limitations of this research concern the data analysis typology. In fact, CCTV is able to provide a graph-based representation of the behavior of the Italian regions that could be extended to different zooms, i.e. by considering the behavior of the individual cities of the Italian regions or by moving the analysis to a wider view, i.e. considering the different European states have faced the pandemic. Thus,  the methodology  analyzes the homogeneous datasets only by applying a statistical test to find similar/dissimilar datasets. Thus, our methodology is able to describe a phenomena. However  the researcher that want to extract a deep knowledge from the data, i.e.  which event  did not cause a region to move from an initial community, or which event that caused a region to move to a more supportive community, for example where fewer  were recorded cases or which interventions may improve the behavior of critical regions, they should recur  to data mining methods.

-spelling mistakes should be elimitated

We apologize for the issue. We fixed typos and mistakes.

Round 2

Reviewer 2 Report

All comments have been addressed.